# Tissue-specific reprogramming of glutamine metabolism maintains tolerance to sepsis

**Brooks P. Leitner**[1,2], **Won D. Lee**[3,4], **Wanling Zhu**[1,2], **Xinyi Zhang**[1,2], **Rafael C. Gaspar**[1,2], **Zongyu Li**[1,2], **Joshua D. Rabinowitz**[3,4,5,6], **Rachel J. Perry**[1,2]*

1 Department of Cellular & Molecular Physiology, Yale University, New Haven, Connecticut, United States of America, 2 Department of Internal Medicine, Yale University, New Haven, Connecticut, United States of America, 3 Lewis Sigler Institute for Integrative Genomics, Princeton University, Princeton, New Jersey, United States of America, 4 Department of Chemistry, Princeton University, Princeton, New Jersey, United States of America, 5 Department of Molecular Biology, Princeton University, Princeton, New Jersey, United States of America, 6 Ludwig Institute for Cancer Research, Princeton Branch, Princeton, New Jersey, United States of America

* rachel.perry@yale.edu

**Data Availability Statement:** All data and code follow the FAIR principle as defined by the NIH Data Commons. Human RNA Transcriptomics can be found at GEO Accession Number GSE13205 at

## Abstract

Reprogramming metabolism is of great therapeutic interest for reducing morbidity and mortality during sepsis-induced critical illness. Disappointing results from randomized controlled trials targeting glutamine and antioxidant metabolism in patients with sepsis have begged a deeper understanding of the tissue-specific metabolic response to sepsis. The current study sought to fill this gap. We analyzed skeletal muscle transcriptomics of critically ill patients, versus elective surgical controls, which revealed reduced expression of genes involved in mitochondrial metabolism and electron transport, with increases in glutathione cycling, glutamine, branched chain, and aromatic amino acid transport. We then performed untargeted metabolomics and $^{13}C$ isotope tracing to analyze systemic and tissue specific metabolic phenotyping in a murine polymicrobial sepsis model. We found an increased number of correlations between the metabolomes of liver, kidney, and spleen, with loss of correlations between the heart and quadriceps and all other organs, pointing to a shared metabolic signature within vital abdominal organs, and unique metabolic signatures for muscles during sepsis. A lowered GSH:GSSG and elevated AMP:ATP ratio in the liver underlie the significant upregulation of isotopically labeled glutamine's contribution to TCA cycle anaplerosis and glutamine-derived glutathione biosynthesis; meanwhile, the skeletal muscle and spleen were the only organs where glutamine's contribution to the TCA cycle was significantly suppressed. These results highlight tissue-specific mitochondrial reprogramming to support liver energetic demands and antioxidant synthesis, rather than global mitochondrial dysfunction, as a metabolic consequence of sepsis.

## Introduction

Sepsis-induced critical illness affects millions of individuals in the US and worldwide every year [1]. Even among survivors, a large fraction experience post-septic complications, with a higher-than-expected proportion of sepsis survivors either dying or returning to hospitals

https://www.ncbi.nlm.nih.gov/geo/. All Python and R code used to create the figures can be found on GitHub at https://github.com/BrooksLeitner/sepsismetabolism. All raw data generated for this study are shown in the figures.

**Funding:** B.P.L is supported under the National Institutes of Health Medical Scientist Training Program Training Grant T32GM007205. W.D.L is supported by NIH grant F32DK127843. This study was supported by unrestricted startup funds from Yale University to R.J.P. The funders played no role in the research or in the decision to publish.

**Competing interests:** The authors have declared that no competing interests exist.

**Abbreviations:** APE, atom percent excess; BCAAs, branched chain amino acids; CLP, cecal ligation and puncture; eWAT, epididymal white adipose tissue; FC, fractional contribution; GSS, glutathione synthase; LC-MS/MS, liquid chromatography-mass spectrometry/mass spectrometry; MDVs, mass distribution vectors; SLC, solute carrier.

within the next year compared to an age-matched population [2, 3]. The acuity of life-threatening sepsis suggests that metabolic modulation may be a promising avenue for therapy. In the past decade, however, numerous trials attempting to supplement critically ill patients with various metabolic modulators, including selenium, glutamine, vitamin C, and thiamine [4–7], have yielded minimal survival benefit. Glutamine supplementation has even been demonstrated as detrimental [8]. This has led critical care leaders of the International Surviving Sepsis Campaign to prioritize a better understanding of the cellular and subcellular mechanisms of sepsis-induced metabolic reprogramming [9].

Sepsis is a condition of disrupted inflammation-initiated metabolic homeostasis. Corresponding to inflammatory response changes, sepsis develops from early (hypermetabolic) to late (hypometabolic) phase. During the early hypermetabolic phase, muscle proteolysis and liver gluconeogenesis increase to provide fuel for fighting against infection and healing wounds. Afterward, during the hypometabolic phase, the body shifts to fatty acid and amino acid oxidation to resist organ damage. Finally, a longer-duration phase follows that corresponds to tolerance to sepsis [10–12]. The hypermetabolic phase is challenging to identify in patients given the heterogeneity of presentation, differences in sepsis onset prior to or while in the intensive care unit or emergency room, in addition to variance in poorly defined patient characteristics [13]. Further, the characterization of damage to muscle and vital organs during the hypometabolic phase is also inadequately understood, with several mechanisms describing the liver's role in coordinating metabolic reprogramming of lipid metabolism and iron metabolism during this time [14–16]. Consequently, there is an urgent need for further understanding of the factors related to systemic metabolic reprogramming during sepsis.

Given that critical illness-induced skeletal muscle wasting has been thoroughly documented, and likely contributes both to mortality and long-term morbidity, we began our investigation with transcriptomic analyses of human skeletal muscle biopsies taken in the intensive care unit. Then, given that evolutionary pressures have required limited wasting of precious amino acids, we reasoned that the constituent amino acids derived from skeletal muscle likely are destined to contribute to other more life-threatening functions. We then performed targeted and untargeted plasma and tissue metabolomics after metabolic characterization of our cecal ligation and puncture (CLP) murine model of polymicrobial sepsis with a comprehensive cytokine panel and metabolic cage studies. Using insight from our discovery-based transcriptomic and metabolomic analyses, we performed stable isotope tracer infusion studies with labelled glutamine and glucose to examine the metabolic mechanisms of sepsis-induced metabolic reprogramming in seven tissues and plasma during sepsis.

The integrative systems approach taken here provides an explanation for the destiny of glutamine during sepsis, and identifies new therapeutic leads for modulating metabolism during sepsis. A better characterization of the shared and tissue-specific metabolic response to sepsis provides greater resolution for metabolism-targeted therapy.

## Results

### Septic patient skeletal muscle transcriptomics reveals electron transport chain dysfunction and upregulation of amino acid transporters

Skeletal muscle biopsies (vastus lateralis) were taken during a previous study in 13 critically ill septic patients currently in the intensive care unit (ICU) and 8 healthy control patients who were undergoing elective surgical procedures with confirmed absence of infection, and the samples were subjected to transcriptomic analysis [17]. Unfortunately, detailed information

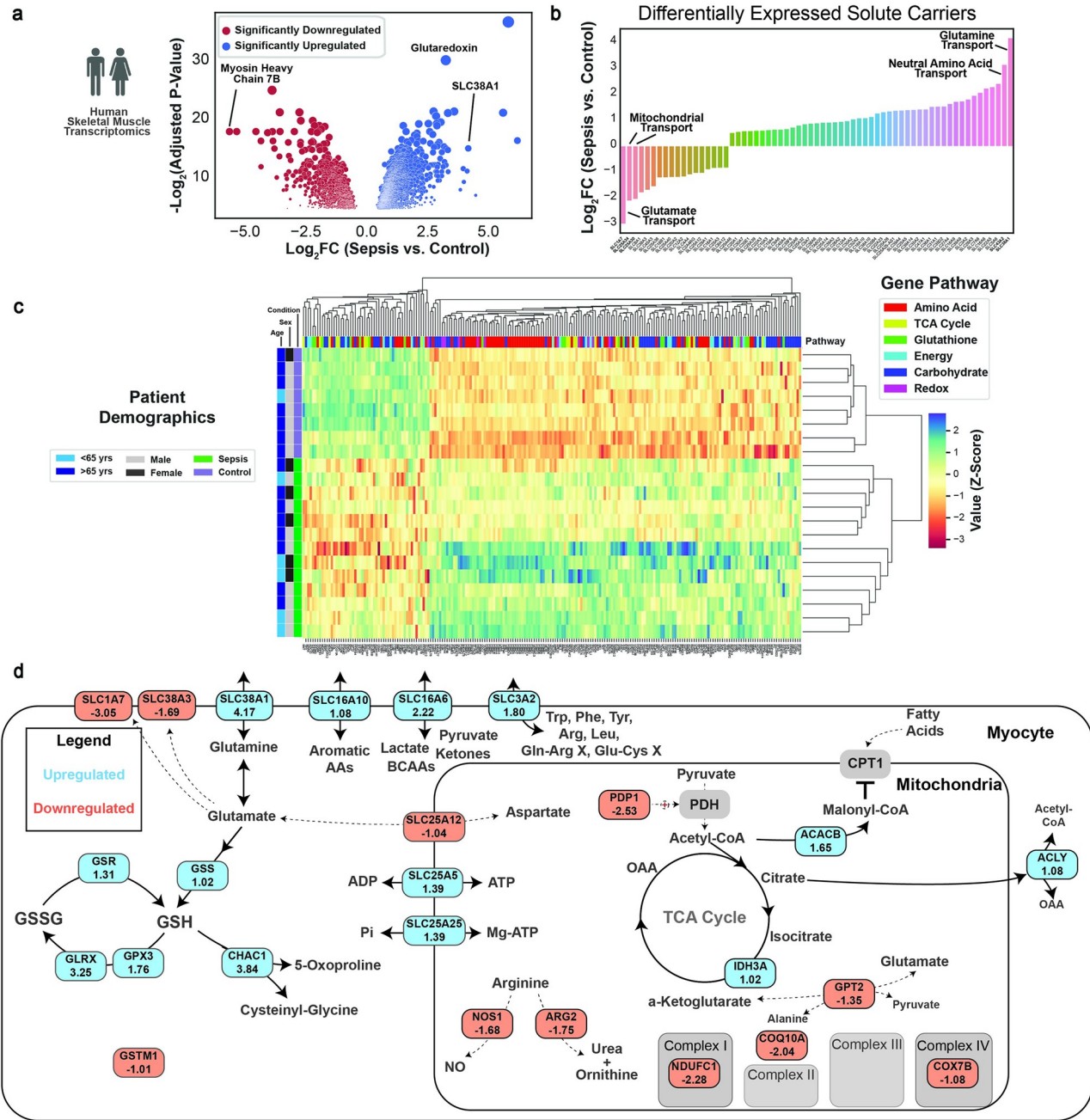

**Fig 1. Human skeletal muscle transcriptomics reveals substantial metabolic reprogramming.** (A) Volcano plot with significantly up and downregulated genes of septic and control patient skeletal muscle gene expression. (B) Barplot of differentially expressed solute carriers. (C) Clustermap of differentially expressed genes in one of six specified pathways. Patient identifiers are on the left most part of the figure, and gene classifiers are at the top portion of the figure. (D) Metabolic map of differentially expressed genes. All displayed genes have an adjusted p<0.05 from differential gene expression analysis.

regarding the duration and severity of sepsis was not available. Thousands of genes were differentially expressed, and notably the gene encoding myosin heavy chain 7B, a structural component of type I oxidative muscle fibers, was downregulated greater than 32-fold compared to control skeletal muscle, suggesting that sepsis induces a major defect in the oxidative muscle

(Fig 1A). Significantly upregulated genes included the glutamine transporter SLC38A1 and glutaredoxin, suggesting reprogramming of glutamine and redox metabolism within skeletal muscle. To characterize metabolites the muscle may prioritize importing or exporting during sepsis, we isolated all solute carrier (SLC) genes (Fig 1B). The most downregulated gene was SLC1A7, the glutamate transporter.

Unsupervised clustering analysis with genes involved in the amino acid, TCA cycle, glutathione, energy, carbohydrate, and redox metabolism again demonstrated clear separation between septic and control patients (Fig 1C, S1 Fig). Patients in the cluster more similar to the control muscle tended to be older, but there was no apparent sex-related clustering.

Focusing on genes that are differentially expressed more than twofold (Fig 1D), we observed significant downregulation of electron transport genes, but upregulation of mitochondrial ATP transport. Notably, mitochondrial GPT2, which mediates glutamate entry into the TCA cycle was downregulated, and glutathione synthase (GSS), and multiple glutathione cycling genes were upregulated. In addition, transporters for branched-chain amino acids (BCAAs), aromatic amino acids, and arginine and cysteine were upregulated, indicating widespread regulation of skeletal muscle amino acid transport. Together, our transcriptomics analysis highlighted a major reprogramming of amino acid handling, and impairment in electron transport chain function in skeletal muscle of sepsis patients.

## Moderate cecal ligation and puncture induces body weight loss, inflammation, and reduction in activity, oxygen consumption, and carbon dioxide production

To further interrogate the metabolic mechanisms of sepsis, we performed a comprehensive characterization of our murine model of polymicrobial sepsis, cecal ligation & puncture (CLP) [18, 19]. We determined that ligation of 0.5cm of cecum (Fig 2A) with a single puncture with a 23g needle, followed by sutures at two locations (Fig 2B and 2C) led to a one-week survival of approximately 68%, which approximates the lower end of survival rate estimates in humans (67–83%) [1, 2] (Fig 2D). On average this degree of CLP induced 20–25% reduction in body weight within 48 hours (Fig 2E). Septic mice also had widespread induction of numerous inflammatory cytokines, including key players IL-6, IL-10, and MCP-1 (Fig 2F). Metabolic cage studies demonstrated a significant reduction in voluntary activity after sepsis induction compared to mice that underwent a sham procedure (Fig 2G), coinciding with reduced water intake (Fig 2H), and reduced oxygen consumption (Fig 2I) and carbon dioxide production (Fig 2J). Based on these data, we performed endpoint metabolic studies at the 16-hour post-sepsis timepoint, where mice are reproducibly hypometabolic: voluntary activity was low and comparable between both sham and CLP mice, and systemic metabolism was substantially reduced in CLP mice.

## Plasma metabolomics reveals systemic metabolic cycle activity, stress hormone activity, and global oxidative stress

Untargeted and targeted liquid chromatography-mass spectrometry/mass spectrometry (LC-MS/MS)-based metabolomics were performed on plasma collected 16 hours post sepsis or sham surgery. Principal component analysis effectively distinguished the experimental groups with principal component 1 explaining 48.5% of the variance in the datasets (Fig 3A). Hippurate, which has been administered as methenamine hippurate to prevent recurrent urinary tract infections [20, 21], was significantly downregulated in sepsis, while the classic inflammation-induced metabolite, itaconate [22], was significantly upregulated (Fig 3B). Metabolites

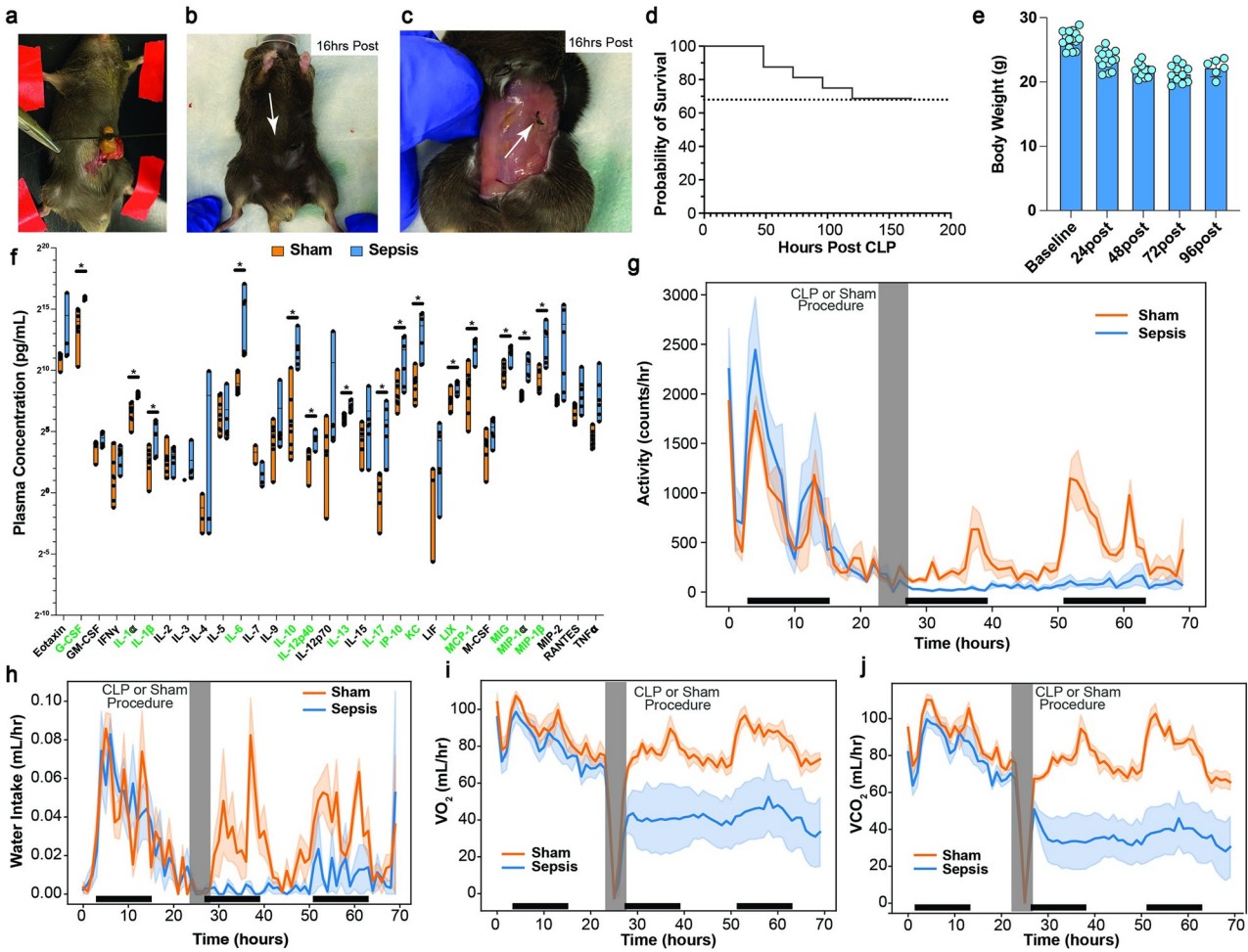

**Fig 2. Murine model of sepsis, cecal ligation and puncture, induces weight and activity loss, widespread inflammation, and significant reduction in energy expenditure.** Photos of degree of ligation (A), superficial suture (B), and internal suture (C) during CLP. (D) Survival curve for degree of mortality. (E) Body weight during CLP challenge. (F) Plasma cytokines in septic and sham control animals. Metabolic cage data showing reduction in activity (G), water intake (H), $VO_2$ (I), and $VCO_2$ (J). Dark bars along the x-axis indicate the animal's dark cycle. Asterisk in F indicates p-value from student's t test < 0.05 between sepsis and sham groups.

within the NAD biosynthetic pathway were significantly disturbed, with a notable 4-fold reduction in circulating nicotinamide and elevation of kynurenine (Fig 3C). Unsupervised clustering analysis revealed significant downregulation of carbohydrate metabolites, with upregulation of glutamine, stress-related metabolites including TCA, urea cycle, and xenobiotic metabolites (Fig 3D). The stress response during the tolerance phase of sepsis was apparent with 4–10 fold increases in tetrahydrocortisol and corticosterone (Fig 3E). Palmitoylcarnitine was upregulated, providing evidence for increased systemic fatty acid oxidation, while deoxycarnitine was downregulated (Fig 3F). Of the amino acids, glutamine elevation and glutamate suppression were the most notable signatures, consistent with the human transcriptomics data (Fig 3G). Consistent with a picture of systemic oxidative stress and mitochondrial dysfunction [23], TCA intermediates were broadly reduced (Fig 3H). Further, the reduced hippurate is consistent with enhanced nitrogen clearance, possibly engaging this particular pathway due to gut microbial synthesis of this metabolite (Fig 3I).

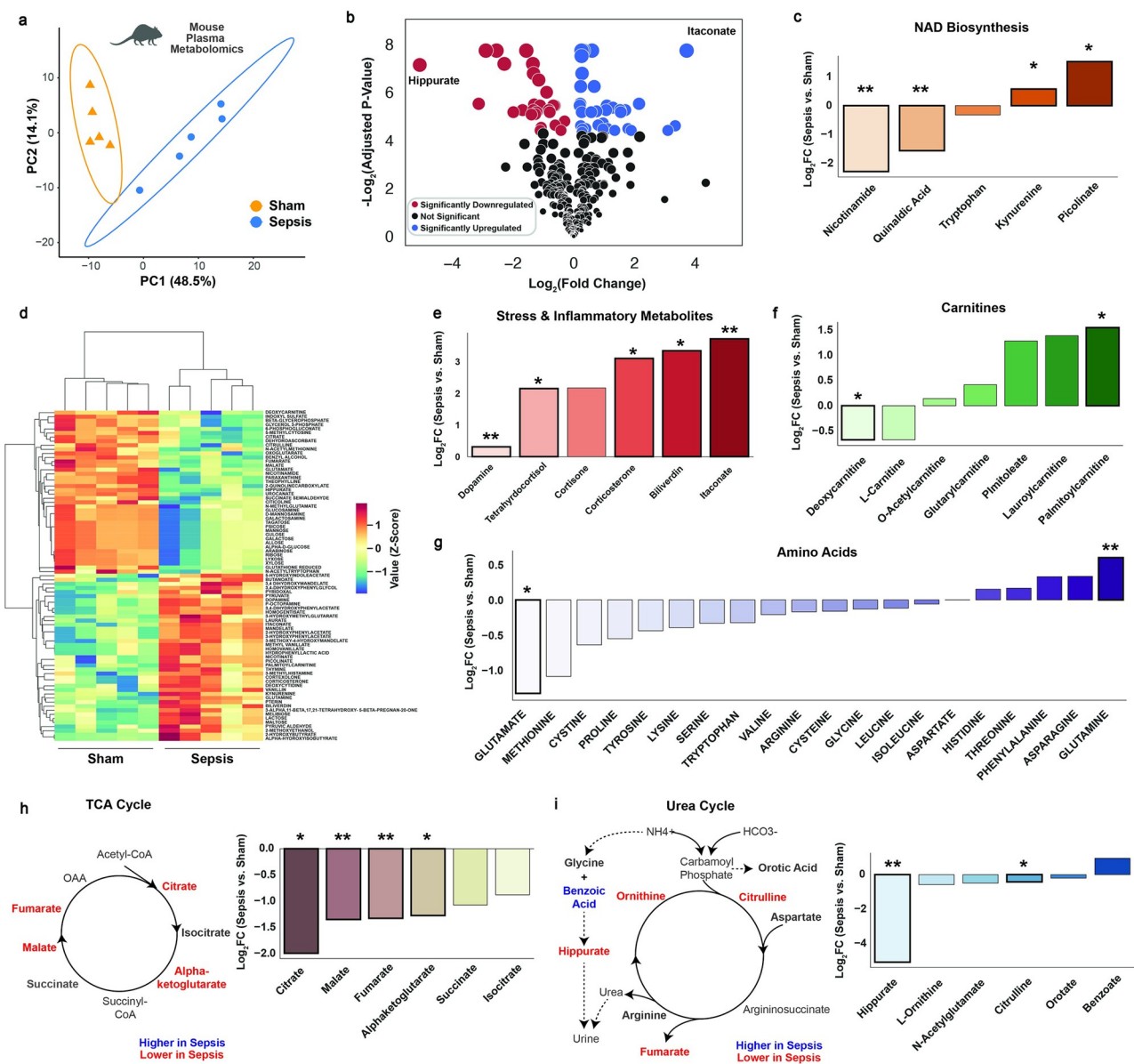

**Fig 3. Plasma metabolomics demonstrates high degree of metabolic reprogramming at the systemic level.** (A) Principal components analysis of plasma metabolomics. (B) Volcano plot, with differentially expressed metabolites (adjusted p<0.05) in red if downregulated, and blue if upregulated in septic mice. (C) Metabolites in the NAD Biosynthetic pathway. (D) Clustermap of all differentially expressed metabolites. Select differentially expressed metabolites related to inflammation/stress (E), carnitines (F), and amino acid metabolism (G). A metabolic map and associated bar plot of key metabolites in the tricarboxylic acid (TCA) cycle and related pathways (H), and in the urea cycle (I). * indicates a benjamani-hochberg adjusted p<0.05, ** indicated <0.01 for septic vs. sham animals.

## Tissue-specific and shared metabolic states are reprogrammed during sepsis

To understand the unique and shared metabolic features of each organ system during pathogen-induced inflammation, we performed unsupervised hierarchical clustering of the metabolites in seven tissues of sham and septic animals (Fig 4A). We found that the same organs were clustered together while sham and septic groups were largely distinguished (particularly in

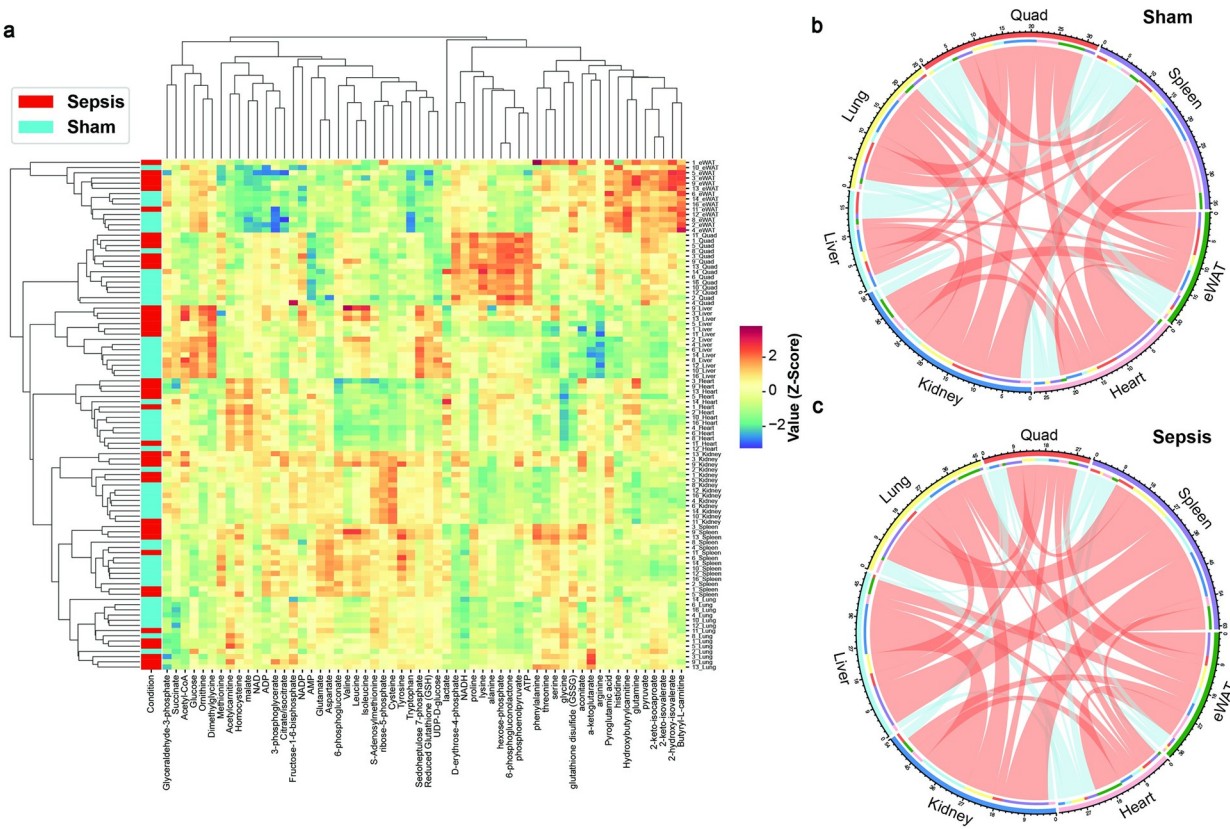

**Fig 4. Shared and tissue-specific metabolic responses to sepsis.** (A) Clustering heatmap shown for all measured intra-tissue metabolites in septic and sham control mice. Chord plot demonstrating significant correlations (p<0.05 for spearman rho correlation) of metabolites between seven tissues in sham (B) and septic (C) mice. Red lines indicate positive correlations, blue lines indicate negative correlations. The line thickness is dictated by the number of metabolites shared between tissues.

liver and skeletal muscle), confirming that tissue-specific and sepsis-specific metabolic features are robust measurements for characterizing the disease in various organs. Interestingly, BCAAs (valine, leucine, and isoleucine) were highly enriched in the liver among other organs, and septic liver had even higher enrichment of these metabolites compared to the sham control. In addition, NADP+ level appeared highest in septic livers, which serves as a key signal for upregulation of NADPH synthetic pathways [24]. There appeared to be a muscle-specific set of metabolites including proline, lysine, alanine, and hexose-phosphate, consistent with both high free amino acid content of skeletal muscle and the lack of glucose-6-phosphatase in skeletal muscle, leading to high intracellular content of glucose-6-phosphate.

To understand the shared metabolic phenotype among different organs, we performed spearman rho correlations between pairs of all tissues (Fig 4B and 4C). In the sham condition, the liver had the fewest (18) significantly correlated metabolites, owing to its unique role in maintaining metabolic homeostasis. The metabolite with the most significant correlations in the control condition was S-adenosyl-methionine (S2A Fig), providing evidence for its ubiquitous role in metabolic homeostasis through one-carbon metabolism at the whole-body level during rest. During sepsis, the relative number of correlations shared by cardiac and skeletal muscle was reduced, while the abdominal organs (spleen, kidney, liver) shared a much greater proportion of metabolites, in particular glutamate and glutamine (S2B Fig). These data suggest

unique metabolic states for heart and skeletal muscle, and a global coordination of key metabolic processes, particularly amino acid metabolism, in other organs in response to sepsis.

## Cellular energy status and cytosolic and mitochondrial redox states set metabolic state in liver and other organs

Next, we investigated whether tissue-specific redox balance and energy states were altered during sepsis. The AMP/ATP ratio, which is a barometer of the energy status of the tissue, was significantly upregulated in the liver, providing a driving force for the influx of metabolites for ATP provision in the liver (Fig 5A). In the liver and spleen, the 2GSH:GSSG ratio was significantly reduced in septic mice, suggesting these organs are particularly under redox stress during sepsis (Fig 5B). At the whole organ level, NAD/NADH ratio was increased only in the septic spleen (S3C Fig). However, cytosolic redox state, indicated by the lactate:pyruvate ratio, was lower in liver, spleen, and epididymal white adipose tissue (eWAT) (Fig 5C). The ADP/ATP ratio was increased modestly in the lung but tended to decrease in the spleen and heart, hinting at the existence of signals to reduce energy production in these two tissues (S3A Fig).

## Glutamine fuels liver glutathione synthesis and TCA-cycle dependent processes, while skeletal muscle glutamine metabolism is reduced

To understand how sepsis impacts systemic nutrient turnover and tissue-specific substrate utilization, we performed stable isotope infusions with $^2H_7$-glucose and $^{13}C_5$-glutamine. Both glucose and glutamine reached steady state enrichments by 110 minutes of the 120-minute infusion (Fig 6A and 6B). We observed that glucose turnover was significantly reduced (Fig 6C) while glutamine turnover was not altered in septic mice (Fig 6D). Despite the lack of difference in whole-body glutamine turnover between sham-operated and CLP mice, we probed further to determine whether any differences in glutamine metabolism may be present. As has been implied [25], circulating glutamine contributed significantly more to *de novo* glutathione synthesis in the liver (Fig 6E), but was not upregulated in any other organs (S4A–S4C Fig). Glutamine also served a more significant role in TCA anaplerosis in the kidney and the liver, as noted by the significant increases of the fractional contribution of glutamine to TCA cycle

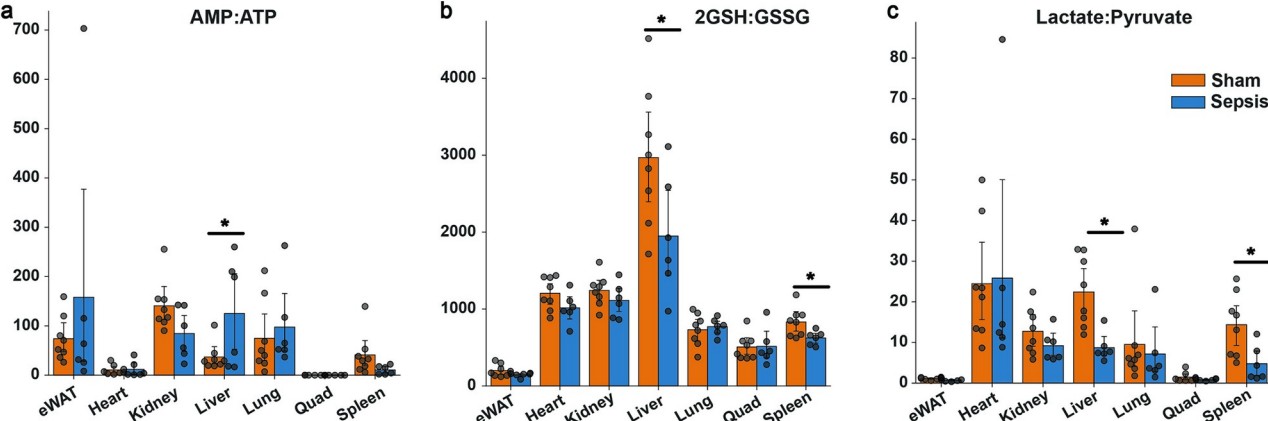

**Fig 5. Sepsis induces global redox and energetic stress in tissues.** (A) Tissue energy state ratios of ADP to ATP and AMP to ATP. (B) Redox ratios of reduced glutathione to glutathione disulfide (2GSH:GSSG) and (C), lactate to pyruvate. Asterisk indicates p<0.05 from a student's t test between septic and sham animals.

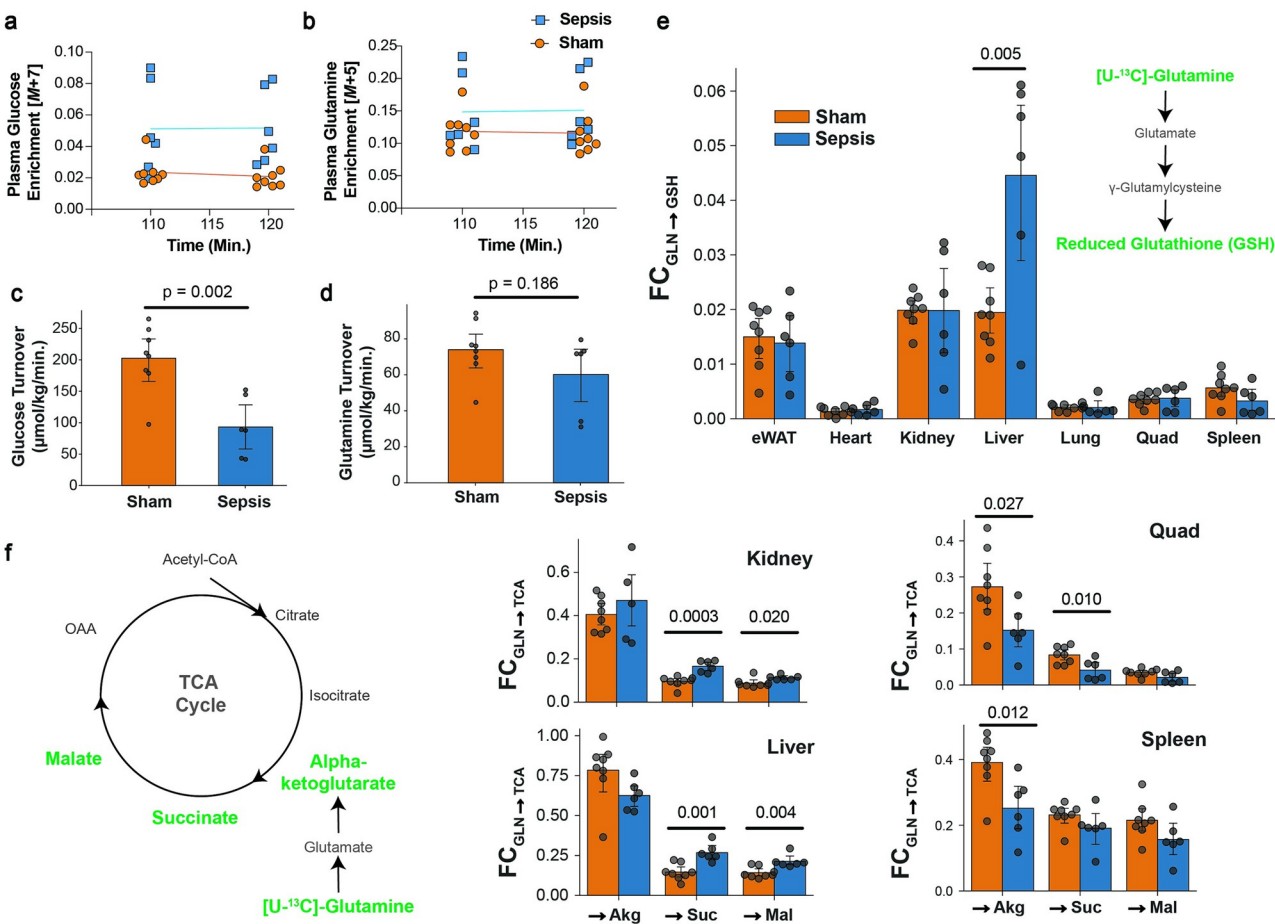

**Fig 6. Stable isotope tracing with U-¹³C₅-Glutamine reveals muscle-liver glutamine reprogramming.** Fully labeled tracer enrichment for [1,2,3,4,5,6,6-²H₇]-glucose (A), and [U-¹³C₅]-glutamine (B) during the final ten minutes of the tracer infusion study. Whole body turnover of glucose (C) and glutamine (D). Fractional contribution of glutamine to glutathione (E), and to three intermediates of the TCA cycle (F). p<0.05 is deemed significant from a student's t test between septic and sham animals.

intermediates (Fig 6F). Meanwhile, glutamine contributed to the TCA cycle to a much lesser degree in the skeletal muscle and the spleen (Fig 6F). Interestingly, the contribution of glutamine to TCA cycle intermediates in liver versus skeletal muscle is opposite to the relative mRNA expression of the glutamine transporter SLC39A1, which is approximately 14-fold higher in skeletal muscle than in liver [26].

## Discussion

The metabolic response to sepsis has long been studied and is still of great interest for guiding therapeutic management of critical illness and long-term morbidity [9]. However, many previously investigated pathways have not led to practice-changing clinical trials, and thus there is a need for data beyond the well-studied metabolic pathways including glucose production [27–30], lactate production [31, 32], and muscle proteolysis [33, 34]. The more recent advent of high-throughput biological omics techniques (transcriptomics, proteomics, metabolomics, and others) provides a timely platform to discover new metabolic mediators of the coordinated metabolomic response to sepsis. Numerous studies have utilized these techniques and

provided unique insights into otherwise underappreciated pathways, most often on samples obtained solely from the plasma and/or from a single organ. Thus, the goal of our study was to elucidate key metabolic pathways across multiple organs that define the metabolic response to sepsis and to determine the metabolic fate of glutamine.

## The tissue specific role of glutamine in whole body metabolism

Glutamine in the context of sepsis-induced critical illness has been investigated thoroughly [5, 6, 35–37], and so far the few consensus opinions are that glutamine supplementation appears ineffective at reducing skeletal muscle proteolysis [38], and that it should not be administered in critically ill patients [39], especially in light of the evidence from the REDOXs randomized controlled trial, in which investigators demonstrated that glutamine supplementation worsened survival in patients with sepsis. Whole body isotope tracing studies have demonstrated that while plasma glutamine and glutamate concentrations go down, whole body production of these metabolites goes up [40]. Our study aimed to contribute to the answer to the following questions: what is the fate of glutamine within the tissue/cell during critical illness? It is well-supported that glutamine is likely derived from skeletal muscle proteolysis [41], but where does it go? A key piece to better understand the intracellular and interorgan role of glutamine during sepsis is the potential separate fates of the glutamine and carbon atoms.

Though we did not directly test the true source of glutamine, it is likely derived from skeletal muscle, the largest reservoir of amino acids in the body. Glutamine and alanine are released from skeletal muscle disproportionately to other amino acids, likely due to their ability to be the final metabolite that carries nitrogen atoms along with carbon precursors from the TCA cycle and the trioses (pyruvate and lactate) respectively [42]. Our data are consistent with the idea that glutamine supplementation does not inhibit proteolysis by uptake and further metabolism, considering substantially lower glutamine incorporation into TCA cycle metabolites in muscle. Thus, as skeletal muscle protein content is a balance of both muscle protein synthesis and proteolysis, it is likely that glutamine derived nitrogen entry into muscle protein synthesis is inhibited while proteolysis is increased. In this manner, all mechanisms are optimized to release skeletal muscle carbon and nitrogen stores.

Much previous work has been done to illustrate that a key fate of glutamine's carbons is to support the carbon backbones for glucose in liver or gut-derived gluconeogenesis [12, 43, 44]. In addition, previous studies have demonstrated that supplemental glutamine helps maintain gut glutathione levels in ischemia reperfusion injury [45], muscle glutathione levels in post-traumatic human muscle [46], and plasma but not liver glutathione in the resting state [47]. Our study provides a comprehensive picture of which tissues upregulate glutamine's contribution to glutathione synthesis and supports a model by which circulating glutamine fuels liver-mediated *de novo* glutathione synthesis, as well as serves as a key anaplerotic substrate in the liver and kidneys. These processes are dependent both on the redox and energy status of the organism. When liver energy demand is high (indicated in our study by an elevated AMP:ATP ratio), substrate delivery can become limiting (typically cysteine for glutathione biosynthesis) for energy and antioxidant-producing pathways.

Though a state of hyperammonemia in sepsis is not a hallmark, it is a sign of poor prognosis [48]. Ammonia and subsequently blood urea nitrogen, are key fates related to glutamine's removal of its two nitrogen atoms as it contributes to cataplerosis and may underlie poor outcomes in sepsis. Though we were unable to track the fate of nitrogen in the current study, it would be important to identify any underlying features that are associated with the inability to clear nitrogen, given that negative nitrogen balance is a common feature of critical illness [49]. Thus, more effective strategies geared toward modulation of glutamine and glutathione

pathways may reconsider targeting liver energy and redox states, more direct inhibition of proteolysis, or clearance of toxic metabolic byproducts of amino acid metabolism (nitrogen species). For example, significant evidence related to the spontaneous mutation in nicotinamide nucleotide transhydrogenase (NNT) gene of C57BL/6 mice has demonstrated a key relationship between mitochondrial redox state, glutathione metabolism, and energy production [50]. All mice in our study were C57BL/6J (not N), but it would be intriguing to interrogate the reduced ability to clear free radical damage, spontaneous oxidation of NADPH, and altered GSH:GSSG ratios in the tissues of mice with the NNT mutation, and further determine if targeting NNT (as has been done in a mouse model of polycystic kidney disease [51]) could be useful in future models of sepsis.

## Sepsis in the metabolic context of other systemic metabolic conditions

Certain metabolic features of sepsis are shared among other various metabolic states, including starvation, ischemia-reperfusion injury (IRI), and several inborn errors of metabolism. In starvation, proteolysis ensues as glycogen stores are depleted, yet proteolysis rates appear to eventually slow [52]. Moreover, at least in human starvation, plasma glutamine is reduced while tyrosine and BCAAs are elevated, while in our experimental sepsis model glutamine was elevated, and BCAAs tended to be reduced. The mechanisms for the shared and distinct regulation of amino acids during sepsis versus starvation remain to be elucidated. In IRI, elevated plasma succinate is a hallmark [53], where we observed reduced succinate, possibly because the putative reduction in perfusion in our study may be less severe than experimental IRI. Our septic metabolome appears nearly opposite that of the mitochondrial disorder leading to mitochondrial encephalomyopathy lactic acidosis and stroke-like episodes (MELAS), which is dominated by reductive, rather than oxidative stress [23]. Though there are shared metabolic features of our experimental model of sepsis and numerous other systemic metabolic conditions, there appear metabolic mechanisms unique to each that need further investigation.

However, the metabolic response to sepsis appears strikingly similar to urea cycle disorders, in particular ornithine transcarbamylase deficiency [54]. Elevated hippurate, which is a renally cleared nitrogen carrier illustrates an intriguing therapeutic target. Benzoic acid is FDA approved for the treatment of urea cycle disorders due to its ability to conjugate with the amino acid glycine and thus serve as a nitrogen carrier in the setting of excess ammonia/urea production. As increased plasma nitrogen content is associated with severity of sepsis [55], enhanced clearance may be an important therapeutic consideration. In addition, targeting nitrogen clearance rather than inhibition or supplementation of specific individual amino acids is plausibly higher yield, as key metabolic process to mount an immune response and maintain homeostasis can be enabled (but likely not enhanced). It remains to be determined whether impaired nitrogen clearance *per se* or something upstream is a main mediator of mortality.

In summary, the metabolic response to sepsis is complex, and a global statement such as "mitochondrial dysfunction" is an over simplistic summarization. We have elucidated significant transcriptomic regulation in human skeletal muscle, related to amino acid transport and glutathione handling. In the hypometabolic phase (characterized by reduced oxygen consumption) of experimental sepsis, we demonstrate that though glucose turnover is substantially reduced, whole-body glutamine turnover remains similar. Our study adds key insights into the fate of glutamine; reduced incorporation into the skeletal muscle TCA cycle, and increased contribution to liver and kidney TCA cycle, and more intriguingly, increased corporation into liver glutathione. These data suggest that tissue-specific metabolic modulation may be required to have an influence on morbidity and mortality, rather than augmentation of a particular metabolic pathway at the whole-body level, raising a new set of challenges in surviving sepsis.

## Materials and methods

### Animals

All studies were approved by the Yale University Institutional Animal Care and Use Committee (protocols 2019 [now 2022]-20290 and 2020–20149). 8-week-old male C57/Bl6J mice were maintained on standard rodent chow diet at room temperature in Yale Animal Research Center maintained housing. Cecal ligation and puncture (CLP) was performed under flow controlled (2.5%) isoflurane anesthesia. A 1 cm left of midline incision was made to both the cutaneous and peritoneal layers. The cecum was isolated and ligated at 0.5cm from the tip, then poked a single time with a 23g needle. Half an mL of PBS was injected into the peritoneal cavity before suturing closed the abdominal wall. Bupivacaine was applied between the abdominal wall and cutaneous skin layers, then a second suture closed the cutaneous portion of the abdomen. Sham control animals had the same procedure without the any manipulation of the cecum. Animals were randomly assigned to sepsis or sham control groups prior to studies. Food was removed from the cages and the mice were singly housed at the time of sham or CLP for survival, body weight, and untargeted plasma metabolomics studies. Mice were euthanized via isoflurane asphyxia followed by cervical dislocation 15–16 hours after CLP for the metabolomics and cytokine assays, and seven days after CLP for body weight and survival studies. The 15–16 hour timepoint was chosen because our metabolic cage data suggested that this timepoint was a period of lowest basal activity and energy expenditure in the sham control mice. We found this timepoint to be the most reproducible post-CLP or sham control procedure that was likely to minimize variability in energy expenditure across study days.

For stable isotope tracing, an intravenous catheter was placed through the jugular vein of the animal under isoflurane anesthesia, then tunneled subcutaneously with a thread at the top of the skin. Mice were singly housed after this surgery and were given carprofen in their drinking water for the following three days and checked daily to ensure recovery after surgery. Six days after catheterization surgery, mice were randomly allocated to CLP or sham control procedures, and food was removed from the cage at the time of surgery. Fourteen hours later, one end of the intravenous catheter was retrieved, flushed with 100 μL of normal saline, then attached to an infusion pump. U-$^{13}$C$_5$-Glutamine and $^2$H$_7$-Glucose were co-infused for two hours (following a 3x prime for five minutes) continuous infusion. Tail blood was collected at minutes 110 and 120, and immediately centrifuged with supernatant plasma collected and stored in a -20˚C freezer, to be used for plasma enrichment and steady state analyses. Then, mice were infused with 50 μL Euthasol (1:10 in PBS) and blood from the inferior vena cava was immediately drawn and tissues collected and frozen in liquid nitrogen cooled Wollenberger tongs. A flow chart depicting the timing of each of these studies can be found in the supplement (Extended Data Fig 5).

### Metabolic cage studies

Eight mice were singly housed three days prior to placement into the Comprehensive Lab Animal Monitoring System apparatus from Columbus Instruments. After 24 hours in the metabolic cages, all mice were removed from the metabolic cages and either sham or CLP was performed within two hours. Mice were returned to the cage for 36 hours post-surgery with body weight determined at the end of the study.

### Cytokine measurement

Blood from the inferior vena cava on the day of the stable isotope tracer studies was spun down for 30 seconds in a centrifuge and placed immediately in a -20C freezer. Samples were

set to Eve Technologies, where the 31-Plex Cytokine Discovery Assay was run according to their instructions.

## Targeted and untargeted plasma metabolomics

Ten mice were randomly subjected to CLP or Sham as described above. 20 μL of plasma was deproteinated and run on two liquid chromatography columns (Hypercarb and Reverse Phase) in both positive and negative modes for a total of 4 sample runs per study sample via liquid chromatography-mass spectrometry/mass spectrometry (LC-MS/MS). Targeted and untargeted metabolite peaks were curated using El-MAVEN software with the IROA library and KEGG database. $^2H_4$-Taurine and $^2H_8$-Phenylalanine were used as internal standards.

## Stable isotope tracing

Sixteen mice were randomly allocated to CLP or Sham as described above. Tracer was diluted in PBS to achieve infusion rates of 5.34 μmol/kg/min for [1,2,3,4,5,6,6-$^2H_7$]-glucose and 10 μmol/kg/min for [U-$^{13}C_5$]-glutamine ~2.5 hours prior to infusion. Mice were weighed immediately prior to infusion and a Harvard Syringe Pump was set to provide the correct infusion rate based on the animal's body weight. Two mice in the septic group did not survive the infusion study, consistent with our survival percentage, and thus there were six mice in the CLP group and eight in the Sham procedure group.

At minutes 110 and 120 blood was drawn from the tail vein via tail massage and collected into capillary tubes, then immediately spun down in heparin-coated tubes, and the plasma was transferred to an Eppendorf for immediate placement into a -20C freezer. Mice were then euthanized and tissues were collected within 75 seconds in the following order after blood was drawn from the inferior vena cava: skeletal muscle, liver, kidney, spleen, epididymal white adipose tissue, heart, lung. Tissues once removed were immediately submerged in liquid nitrogen after clamping with pre-cooled Wollenberger clamps.

Tissues were ground to a fine powder under liquid nitrogen with a mortar and pestle. Ground tissue was then weighed (∼20 mg) and mixed with −20˚C 40:40:20 methanol:acetonitrile:water (extraction solvent) at a concentration of 25 mg/mL. Extract was then vortexed and centrifuged twice at 16,000 x g for 20 min at 4˚C before the final supernatant was transferred to LC-MS tubes for analysis.

## Targeted metabolite measurement with enrichment by LC-MS/MS

The mass spectrometry analysis of polar metabolites was performed using Orbitrap Exploris 480 mass spectrometer (Thermo Fisher Scientific) coupled with hydrophilic interaction chromatography (HILIC). An XBridge BEH Amide column (150 mm × 2.1 mm, 2.5 μM particle size, Waters, Milford, MA) was used. The gradient was solvent A (95%:5% H2O:acetonitrile with 20 mM ammonium acetate, 20 mM ammonium hydroxide, pH 9.4) and solvent B (100% acetonitrile) 0 min, 90% B; 2 min, 90% B; 3 min, 75%; 7 min, 75% B; 8 min, 70% B, 9 min, 70% B; 10 min, 50% B; 12 min, 50% B; 13 min, 25% B; 14 min, 25% B; 16 min, 0% B, 20.5 min, 0% B; 21 min, 90% B; 25 min, 90% B. The flow rate was 150 μL/min with an injection volume of 5 μL and a column temperature of 25˚C. The mass spectrometry scans were in negative ion mode with a resolution of 480,000 at m/z 200. The automatic gain control (AGC) target was $1 \times 10^6$ and the scan range was m/z 70–1000.

Data were analyzed using the El-MAVEN (Elucidata) software [56]. Natural $^{13}C$ and $^2H$ abundances were corrected using the AccuCor2 package in R [57].

## Human skeletal muscle transcriptomics

As stated in the original paper reporting these data [17], the study protocol conformed to the ethical guidelines of the 1975 declaration of Helsinki and had received an *a priori* approval by the Ethical committee of Karolinska Institutet, Stockholm, Sweden and local ethics approval for gene expression analysis at Heriot-Watt University, Edinburgh, Scotland. Data were accessed from GSE13205 using GEOquery package in R. All available data were analyzed. The authors had no access to any information that could identify individual participants during or after data collection. Genes were filtered for low counts, then differential gene expression analysis was performed using Limma. Principal components analysis was performed in R, and then the differentially expressed gene list (all values with a benjamani-hochberg corrected p-value of less than 0.05) were exported to a CSV file. Gene set enrichment analysis was performed using EnrichR [58], and HumanCyc 2016 pathways were selected due to their emphasis on metabolic pathways.

The volcano plot, bar plot, and clustering heatmap were created and generated in Python using the Seaborn package. The significantly differentially expressed genes were filtered using gene lists related to amino acid metabolism, the TCA cycle, glutathione metabolism, energy metabolism, carbohydrate metabolic and redox pathways. These gene lists were obtained from Reactome and MSigDB [59, 60].

## Plasma and tissue isotope tracing

Metabolite peaks were background corrected using AccuCor2 [57], then converted to mass distribution vectors, such that each isotopologue was set as a fraction of the sum of the total isotopologues for each molecule. Pool size measurements, which reflect total molecular peak abundance, were used for calculating redox and energy status ratios. Student's t tests were performed for CLP vs. Sham in GraphPad Prism, with a p-value $<0.05$ considered significant.

Turnover was calculated using the tracer to sum ratio method [61]. Binomial expansions were used to obtain natural abundance and tracer enrichment (98% for glucose, 99% for glutamine) mass distribution vectors (MDVs) of deuterium (0.0156%) and of $^{13}$C (1.1%) in glutamine [62, 63].

$$F(Mx) = b(x; n, p) = \begin{bmatrix} n \\ x \end{bmatrix} (p)^x (1-p)^{n-x}$$

where F is the frequency of an isotopologues, n is the number of potentially labeled atoms in the molecule, x is the number of actually labeled atoms in the molecule, and p is the probability of there being label in the atom (which corresponds either to the tracer enrichment or natural abundance enrichment). We devised a Python function that takes the MDVs of natural abundance (background), tracer, and mixed sample (plasma) enrichments, and computes a linear regression that resolves the estimate contribution from all three inputs to calculate turnover, which is available at GitHub: https://github.com/BrooksLeitner/sepsismetabolism).

As we infused with a highly deuterated compound, which may be subject to primary and secondary isotope effects [64–67], only $^{13}$C-labeled isotopologues were used for each metabolite in tissues. This method also minimizes the confounding influence of secondary tracers due to deuterium incorporation after metabolite flux through various pathways.

Fractional contribution (FC) analyses were performed using uniformly $^{13}$C-labeled plasma glutamine enrichment as the assumed only $^{13}$C labelled precursor. Atom percent excess (APE) was calculated as described [68, 69] from MDVs of plasma glutamine, and each specified tissue

metabolite.

$$APE\left(\overrightarrow{X}\right) = \frac{1}{N}\sum_{j=0}^{N} j\, X_j = \frac{X_1 + 2X_2 + ... + NX_N}{N}$$

where $\overrightarrow{X}$ is the MDV of a molecule, N is the number of potentially labelled atoms in the molecule, and X is the isotopologue. This calculation of APE takes into consideration all labelled isotopologues of each metabolite and provides a weight that increases as the isotopologues is more highly labeled, and then is taken relative to the number of carbons, and also accounts for *in vivo* label dilution. For example, using the precursor-product relationship, $FC_{Gln \to GSH}$ described the calculation of (Tissue Glutathione APE) / (Plasma Glutamine APE). We devised a Python program that calculates APE for any metabolite given its MDV, which is available at GitHub: https://github.com/BrooksLeitner/sepsismetabolism).

## Statistical analysis

Where comparisons between two groups were performed, student's t tests were used to compare, with a p value $< 0.05$ used to determine statistical significance. All No specific methods were utilized to address potential bias; however, all analyses were carried out and monitored by at least two investigators.

**Human skeletal muscle transcriptomics.**   The statistical methods are included in the "Human Skeletal Muscle Transcriptomics" section above.

**Metabolic cage studies.**   A running filter down sampled the data to every hour and were plotted as means and SEMs using the Seaborn Python package.

**Cytokine measurement.**   Cytokines were quantified by Eve Technologies (Details in the Key Resources Table) 31-Plex Cytokine Discovery Assay, and a student's t-test was used to compare sepsis vs. sham, with a p-value $<0.05$ deemed significant.

**Plasma metabolomics.**   2,531 unique metabolites were annotated. The fraction of each metabolite to the sum of all metabolites was computed for each metabolite, then $\log_2$(fraction of total) values were computed. Student's t tests were performed on all metabolites between CLP and Sham, with a Benjamani-Hochberg p-value correction applied in Python (Code is available at GitHub: https://github.com/BrooksLeitner/sepsismetabolism). Differentially expressed metabolites were defined as an adjusted p-value $<0.05$, with no logFC threshold, as these analyses were used as a discovery analysis.

The PCA plot was generated in R. Pathway enrichment analysis was performed using the MetaboAnalyst V5 [70] web tool. The volcano plot, bar plots, and clustering heatmap were created and generated in Python using the Seaborn package.

## Supporting information

**S1 Fig. Principal components analysis of human skeletal muscle transcriptomics in patients with sepsis or control patients.**
(TIF)

**S2 Fig. The cumulative number of significant correlations of intra-tissue metabolites between tissues in sham control mice (a), and septic mice that underwent cecal ligation and puncture (CLP).**
(TIF)

**S3 Fig. Intratissue redox and energetic ratios within tissues of mice with and without sepsis.** Blue bars are septic mice and orange bars are sham control mice. *p $< 0.05$ by student's t-

test.
(TIF)

**S4 Fig. Fractional contribution of uniformly labeled $^{13}$C glutamine into tricarboxylic acid cycle (TCA) intermediates in the (a) heart, (b) lung, and (c) epidydimal white adipose tissue (eWAT).** Akg = alphaketoglutarate, Suc = succinate, Mal = malate, GLN = glutamine. Blue bars are septic mice and orange bars are sham control mice.
(TIF)

**S5 Fig. Flow chart of timing related to experiments.** The timeline is given in hours relative to the CLP or Sham procedure (Time 0). The metabolic cage studies were conducted for the 24 hours before, and 48 hours after the procedure. The untargeted metabolomics and cytokine assays were collected 16 hours post-sepsis from the inferior vena cava (IVC). The targeted tissue metabolomics and isotope tracer experiments were performed at the same time, following a 2 hour isotope infusion, ending between 16–17 hours post procedure. Blue bars indicate approximate timing of animal involvement of data collection.
(TIF)

## Acknowledgments

We gratefully acknowledge Dr. Gang Peng for conversations on statistical analyses, Ali Nasiri for assistance with metabolic cage studies, and the staff at the Yale Animal Resources Center for their careful monitoring of the animals in this study. In addition, we thank Dr. Andrew Wang for guidance on performing cecal ligation and puncture. We thank the IOMIC Core at Yale for running the untargeted metabolomics. We are grateful to all members of the Perry and Rabinowitz labs for their helpful discussions.

## Author Contributions

**Conceptualization:** Brooks P. Leitner, Rachel J. Perry.

**Funding acquisition:** Joshua D. Rabinowitz, Rachel J. Perry.

**Investigation:** Brooks P. Leitner, Won D. Lee, Wanling Zhu, Xinyi Zhang, Rafael C. Gaspar, Zongyu Li.

**Methodology:** Brooks P. Leitner, Won D. Lee, Joshua D. Rabinowitz, Rachel J. Perry.

**Resources:** Joshua D. Rabinowitz, Rachel J. Perry.

**Supervision:** Rachel J. Perry.

**Writing – original draft:** Brooks P. Leitner.

**Writing – review & editing:** Brooks P. Leitner, Won D. Lee, Joshua D. Rabinowitz, Rachel J. Perry.

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
