## [Decision Letter · Decision Letter 0]

25 Apr 2023

PONE-D-23-06470Tissue-Specific Reprogramming of Glutamine Metabolism Maintains Tolerance to SepsisPLOS ONE

Dear Dr. Perry,

Thank you for submitting your manuscript to PLOS ONE. After careful consideration, we feel that it has merit but does not fully meet PLOS ONE’s publication criteria as it currently stands. Therefore, we invite you to submit a revised version of the manuscript that addresses the points raised during the review process.

We look forward to receiving your revised manuscript.

Kind regards,

Juan J Loor

Academic Editor

PLOS ONE

Journal Requirements:

Reviewers' comments:

Reviewer's Responses to Questions

**Comments to the Author**

1. Is the manuscript technically sound, and do the data support the conclusions?

Reviewer #1: Yes

Reviewer #2: Yes

2. Has the statistical analysis been performed appropriately and rigorously? 

Reviewer #1: N/A

Reviewer #2: Yes

3. Have the authors made all data underlying the findings in their manuscript fully available?

Reviewer #1: Yes

Reviewer #2: Yes

4. Is the manuscript presented in an intelligible fashion and written in standard English?

Reviewer #1: Yes

Reviewer #2: Yes

5. Review Comments to the Author

Reviewer #1: The paper describes in a almost qualitive way part of the changes in glutamine metabolism in sepsis. The results are interesting, albeit that there are some technical issues that need to be addressed. Also, the number of observations is rather small. The paper need to be brough in perspective of what is known from human research. The mouse model is probably not a good model of human sepssi.

Specific comments

#102: Already 13 year old study. Indicate why severity was not available. Every ICU at least collect Apache or alike.

#103-126. Several recent studies show that in humans, the glutamine and glutamate production is substantially increased. Do you have data to support that in muscle?

#151: I think hat you should have studied the mice at 24h or even better 48h to make it a more comparable model to human sepsis

#178: In humans, plasma glutamine concentration goes down.

#225: I would show some of the tissue glutamine concentrations as it is the focus of the paper

#263: A primed constant infusion of glutamine does not lead to a plasma glutamine enrichment steady state. The fact that only 110 and 120 min is measured, makes it impossible to establish steady state. Please check onther research in this field. In humans, a very large increase of glutamine production is observed, using a pulse approach that has not this problem of steady state.

#286: Please use subheaders. Discussion is difficult to read.

#306-309: This is known for many years. Please read literature better. The largest part is taken up by liver and even less taken up by gut.

#309-311: I think you need to differentiate what happens with the C and N atoms of glutamine. That are different pathways

#317-318: That is a bold statement and I am not sure you have the data to make that argument. For instance, did you measure proteolysis?

#320: This I also do not understand. Many, many studies show increased muscle protein breakdown in sepsis. So what are you referring too?

#455: Did you use enriched (internal) standard to be able to get the true enrichment?

#520: So you did not do any stats? I actually found several P values in figures. Show all individual data points when possible in all figures and use mean or geomean with 95% CI.

Figure 2e: Do you show mean with 95% CI here?

Figure 2f: Show individual data points like you did in 2e

Figure 3c, e, f, g, g, i: Show individual data points like you did in 2e

Figure 5: Individual points with 95% CI

Figure 6b: Very high glutamine enrichment. Not really a tracer anymore

Figure 6: Individual points with 95% CI

Reviewer #2: The manuscript design is reasonable and the data is reliable, which provides some new insights into the metabolic change and mechanism analysis of glutamine metabolism in skeletal muscles of sepsis patients. However, there are some shortcomings that need to be addressed:

1. The study used multiple experimental methods. If a flowchart could be used to include information such as animal model preparation, isotope labeling, and sampling and detection time points during the experimental process, it would be beneficial for readers to clearly understand the experimental process.

2. In the metabolite LC-MS measurement, why was only the negative ion scanning mode used and the positive ion mode ignored?

3. In the data of the metabolism cage (Figure 2G), there seems to be a problem with the activity data of the animals. The activity data of the animals in the sham group should be similar every day, so why is there such a large difference at different times of detection? On the other hand, the water intake and oxygen consumption of the animals were relatively consistent. How can this be explained?

4. Whether the oxidative stress and mitochondrial dysfunction described in the paper results are caused by Nnt mutation needs to be further clarified.

5. The author used the GEO database to retrieve the gene chip expression dataset (GSE13205) of sepsis patients, analyzed 21 samples, including 13 sepsis patients and 8 non-sepsis patients, and the tissues were all pathological specimens of skeletal muscle fibers. Through pathway enrichment analysis of differentially expressed genes, it was found that there was metabolic reprogramming of glutamine in skeletal muscles of sepsis patients. If the detection data of skeletal muscle proteomics and metabolic pathway-related enzyme activity could be added, the conclusions of the paper will be more convincing.

6. PLOS authors have the option to publish the peer review history of their article (what does this mean?). If published, this will include your full peer review and any attached files.

Reviewer #1: No

Reviewer #2: No

---

## [Author Response · Author response to Decision Letter 0]

2 May 2023

We thank the reviewers for their careful and constructive read of our manuscript, and for the suggestions for improvement that they provided. We have revised accordingly, and hope the reviewers and editors agree with our team that the revised manuscript is now substantially improved. Our responses to each comment are shown below (in bold in the rebuttal document).

Reviewer #1: The paper describes in a almost qualitive way part of the changes in glutamine metabolism in sepsis. The results are interesting, albeit that there are some technical issues that need to be addressed. Also, the number of observations is rather small. The paper need to be brough in perspective of what is known from human research. The mouse model is probably not a good model of human sepssi.

Specific comments

#102: Already 13 year old study. Indicate why severity was not available. Every ICU at least collect Apache or alike.

We believe that a nutrient metabolism-oriented reanalysis of the dataset (GSE13205) allows us to provide insights that would help frame our current study. All data that were available at the GEO website for this dataset included each individual’s accession number, sex, age, and designation of septic versus control. We agree that more detailed information, including a sepsis severity score, would be useful for deeper interpretation of our findings. Unfortunately, these data were not available in this retrospective study in the GEO website.

#103-126. Several recent studies show that in humans, the glutamine and glutamate production is substantially increased. Do you have data to support that in muscle?

We agree that there are many interesting studies that examine the role of amino acid reprogramming during sepsis, including the very recent study demonstrating the pulse-administration derived finding of increase glutamine and glutamate whole-body production (doi: 10.1016/j.metabol.2023.155400). A considerable weakness of using gene expression for the study of metabolism is that it is very difficult to correlate changes in RNA content with metabolite production. In the human RNA expression data, we are not able to directly support or refute the production of amino acids (including glutamine or glutamate) in human skeletal muscle. 

Our isotope tracing studies actually demonstrate a decreased contribution of glutamine and glutamate to the TCA cycle and glutathione biosynthesis, but we did not observe changes in absolute concentrations of metabolites. Obtaining repeat muscle samples in septic mice may induce such a significant stress as we found it important for the animals to be awake and not under anesthesia during the physiological measurements obtained during sepsis, considering that anesthesia is a well-known inducer of metabolic changes likely resulting from stress (e.g. PMID 31672472, 27802272). 

#151: I think hat you should have studied the mice at 24h or even better 48h to make it a more comparable model to human sepsis

Choosing the ideal time post-procedure was challenging for us, and we understand limitations regardless of the ideal post-procedure timepoint chosen. We chose the 16-hour timepoint for several reasons, both from a data-driven and theoretically translational viewpoint.

1. We wanted to choose a timepoint post-CLP or sham in which the basal metabolism of the sham group was most comparable to the CLP group. As shown in Figures 2g-j, we found that the sham group’s activity, water intake, VO2, and VCO2 appeared to be minimal between 14-18 hours post procedure. We thought that this point of comparison would thus be the most reproducible timepoint of comparison between groups.

2. The longer the period post procedure, the more challenging it became to match food and water intake between sham control and CLP mice. We aimed to perform the major metabolic analyses within one full day’s feeding cycle to minimize disruption of this confounding variable.

3. Given that mouse whole body metabolic rates are faster than human metabolic rates, we believe that 16 hours may be comparable to the 3-5 day incubation period from point of bacteremia to severe sepsis in humans. 

We have added a point related to this in the text:

“The 15-16 hour timepoint was chosen because our metabolic cage data suggested that this timepoint was a period of lowest basal activity and energy expenditure in the sham control mice. We found this timepoint to be the most reproducible post-CLP or sham control procedure that was likely to minimize variability in energy expenditure across study days.”

#178: In humans, plasma glutamine concentration goes down.

This statement (that we agree is supported by data including Deutz et al 2021 Clinical Nutrition, though in this study the ICU group had significantly higher lean body mass and lower fat mass) is the impetus for the failed REDOXS 2013 Trial in the NEJM where glutamine supplementation (for the thought that hypoglutaminemia is a hallmark of poor sepsis outcomes) proved to be harmful in patients with critical illness. However, as you know this data has been recently called into question given the disappointing result of this study and other interventional studies. A nice review by Smedberg and Wernerman in Critical Care (Figure 1) DOI: 10.1186/s13054-016-1531-y (among other even more recent papers) put into context that high or low plasma glutamine levels have each been associated with poor outcomes during sepsis.

Nevertheless, we have included this point in our discussion and cited an appropriate paper: “Isotope tracing studies have demonstrated that while plasma glutamine and glutamate concentrations go down, whole body production of these metabolites goes up (Deutz et al 2021 Clinical Nutrition).”

#225: I would show some of the tissue glutamine concentrations as it is the focus of the paper

Our metabolomics method only allowed for relative quantitation, rather than absolute quantitation to show tissue glutamine concentrations. 

#263: A primed constant infusion of glutamine does not lead to a plasma glutamine enrichment steady state. The fact that only 110 and 120 min is measured, makes it impossible to establish steady state. Please check onther research in this field. In humans, a very large increase of glutamine production is observed, using a pulse approach that has not this problem of steady state.

We agree that our approach with a primed constant infusion has limitations, and it will always be extremely challenging to obtain both isotopic and physiological steady state in an experiment as stressful as sepsis. Given that 110 and 120 minutes were within the same range leading up to euthanasia at minute ~121 is at least suggestive of approximating isotopic steady state. Given the limited blood volume of a critically ill mouse, we had to choose our time points carefully to minimize blood loss and further a hypovolemic stress on the rodents.

As the reviewer recognizes, a key aspect of high-quality pulse chase isotopic experiments requires frequent blood sampling to obtain reliable decay curves. This experimental design is infeasible in 20-25g mice without inducing a significant volume or hemoglobin depletion stress on top of sepsis. 

#286: Please use subheaders. Discussion is difficult to read.

Subheaders have been added.

#306-309: This is known for many years. Please read literature better. The largest part is taken up by liver and even less taken up by gut.

Citations have been added that suggest that glutamine is taken up by the liver and gut for gluconeogenesis (Meinz et al 1998, Newsholme and Carrie 1994, Vary et al 1989). The fate of the carbons in these tissues, however, had not been well characterized. Even studies that have shown that glutamine supplementation alters concentrations of glutathione do not demonstrate in which tissues the transfer of glutamine carbons occurs.

A significant addition to the discussion has been added with additional (n=10) citations. 

#309-311: I think you need to differentiate what happens with the C and N atoms of glutamine. That are different pathways

A new section in the discussion has been added to address this important question in the context of prior literature. 

#317-318: That is a bold statement and I am not sure you have the data to make that argument. For instance, did you measure proteolysis?

We have changed this statement from “support” to “are consistent with” in order to make the statement less bold.

#320: This I also do not understand. Many, many studies show increased muscle protein breakdown in sepsis. So what are you referring too?

We are referring to the fact that skeletal muscle protein homeostasis is a balance of muscle protein synthesis and muscle protein breakdown, driven by different mechanistic processes. We have revised the statement to better reflect that our idea is that muscle protein synthesis is completely inhibited during sepsis (in addition to accelerated proteolysis):

“Thus, as skeletal muscle protein content is a balance of both muscle protein synthesis and proteolysis, it is likely that glutamine derived nitrogen entry into muscle protein synthesis is inhibited at the same time that proteolysis is increased. In this manner, all mechanisms are optimized to release skeletal muscle amino acid stores.”

#455: Did you use enriched (internal) standard to be able to get the true enrichment?

We understand and agree with the reviewer’s implication that, to compensate for matrix effects, ion suppression and/or ion enhancement, labeled internal standard can be used. However, we reasoned that introducing a labeled internal standard from a metabolite already labeled in our tracer study could create more confounding effects on the isotopomers than it would clarify; thus we have followed the commonly accepted, standard protocols of our collaborators in the Rabinowitz lab to measure enrichment (e.g. PMID 37100997, 36725930, 36476934, 36223763, 36055202, 35425930, 35058631, 34845393, 34799699, 34588310, 34559996, 34239352, 34223403, and others).

#520: So you did not do any stats? I actually found several P values in figures. Show all individual data points when possible in all figures and use mean or geomean with 95% CI.

All relevant figures have been changed to show individual data points with 95% CI.

Figure 2e: Do you show mean with 95% CI here?

The figure has been changed to show the mean with 95% CI.

Figure 2f: Show individual data points like you did in 2e

Individual points are shown now.

Figure 3c, e, f, g, g, i: Show individual data points like you did in 2e

Log2FC were computed for the untargeted metabolomics analyses, which are calculated based on averages at the group level. Individual points are shown in all other relevant analyses. 

Figure 5: Individual points with 95% CI

The figures have been changed to show individual data points with 95% CI

Figure 6b: Very high glutamine enrichment. Not really a tracer anymore

We agree that the glutamine enrichment was high relative to many tracer studies, which we aimed to account for by adjusting the infusion rate in the septic mice, but was challenging to achieve given the heterogeneity of the CLP procedure. However a recent study has demonstrated that U13C-Glutamine enrichment as high as 15% does not impact whole body glutamine turnover, nor fluxes entering the TCA cycle (Hubbard et al Cell Metabolism Fig S1A and Fig S1B, https://doi.org/10.1016/j.cmet.2022.11.011). 

Figure 6: Individual points with 95% CI

The figures have been changed to show individual data points with 95% CI

Reviewer #2: The manuscript design is reasonable and the data is reliable, which provides some new insights into the metabolic change and mechanism analysis of glutamine metabolism in skeletal muscles of sepsis patients. However, there are some shortcomings that need to be addressed:

1. The study used multiple experimental methods. If a flowchart could be used to include information such as animal model preparation, isotope labeling, and sampling and detection time points during the experimental process, it would be beneficial for readers to clearly understand the experimental process.

We thank the reviewer for considering “the manuscript design is reasonable and the data is reliable.” A supplemental figure with a flowchart was included to depict when each sample/data was collected in relation to the CLP/Sham Procedure.

2. In the metabolite LC-MS measurement, why was only the negative ion scanning mode used and the positive ion mode ignored?

For the untargeted LC-MS plasma metabolomics, metabolites detected in BOTH positive and negative ion mode were used. These details were written in the text under the header “Plasma Metabolomics.”

We used two different machines and methods to detect the labeled metabolites (which a negative ion mode protocol has been optimized to reproducibly measure a targeted set of atom-specific enrichment), and the untargeted metabolites.

The header for Plasma Metabolomics has been changed to “Targeted and Untargeted Plasma Metabolomics” and the second section has been modified to “Targeted Metabolite Measurement with Enrichment by LC-MS/MS” to reflect this distinction more clearly.

3. In the data of the metabolism cage (Figure 2G), there seems to be a problem with the activity data of the animals. The activity data of the animals in the sham group should be similar every day, so why is there such a large difference at different times of detection? On the other hand, the water intake and oxygen consumption of the animals were relatively consistent. How can this be explained?

We believe that the variability in the metabolic cage data may be explained by the inherent heterogeneity of a CLP sepsis and sham procedure under anesthesia. The control animals reduced their activity and water intake in the 24 hours following the sham procedure, which we believe is normal behavior given they still underwent a traumatic and invasive procedure under general, then local anesthesia. Regarding the oxygen consumption data not changing to a significant degree, this is likely due to the fact that activity thermogenesis and the thermic effect of food make up only a small fraction of a mouse’s whole body energy expenditure. Even large reductions in activity may result in small reductions in oxygen consumption and CO2 production in the sham control mice. 

4. Whether the oxidative stress and mitochondrial dysfunction described in the paper results are caused by Nnt mutation needs to be further clarified.

A section in the discussion has been added to address NNT’s role in redox and energy metabolism, and directs the reader to a novel conceptual target of NNT used for ADPKD, which may be modified if relevant in critical illness or other states of significant oxidative stress. 

5. The author used the GEO database to retrieve the gene chip expression dataset (GSE13205) of sepsis patients, analyzed 21 samples, including 13 sepsis patients and 8 non-sepsis patients, and the tissues were all pathological specimens of skeletal muscle fibers. Through pathway enrichment analysis of differentially expressed genes, it was found that there was metabolic reprogramming of glutamine in skeletal muscles of sepsis patients. If the detection data of skeletal muscle proteomics and metabolic pathway-related enzyme activity could be added, the conclusions of the paper will be more convincing.

We agree with the reviewer. A considerable weakness of using gene expression for the study of metabolism is that it is very difficult to correlate changes in RNA content with metabolite production. Though protein content would not alone be able to help determine metabolic flux through tissues, protein content and enzymatic activity would allow for much greater insights into the intra-tissue metabolic mechanisms during sepsis. Unfortunately, this data does not exist in the same cohort of patients where we found the RNA expression. Thus, we respectfully put forth that the combination of transcriptomics with flux analysis in this manuscript generates a data allowing insights into the underlying mechanisms and effects of sepsis-induced metabolic (patho)physiology.

---

## [Decision Letter · Decision Letter 1]

18 May 2023

Tissue-Specific Reprogramming of Glutamine Metabolism Maintains Tolerance to Sepsis

PONE-D-23-06470R1

Dear Dr. Perry,

We’re pleased to inform you that your manuscript has been judged scientifically suitable for publication and will be formally accepted for publication once it meets all outstanding technical requirements.

Kind regards,

Juan J Loor

Academic Editor

PLOS ONE

Additional Editor Comments (optional):

Reviewers' comments:

Reviewer's Responses to Questions

**Comments to the Author**

1. If the authors have adequately addressed your comments raised in a previous round of review and you feel that this manuscript is now acceptable for publication, you may indicate that here to bypass the “Comments to the Author” section, enter your conflict of interest statement in the “Confidential to Editor” section, and submit your "Accept" recommendation.

Reviewer #1: All comments have been addressed

Reviewer #2: (No Response)

2. Is the manuscript technically sound, and do the data support the conclusions?

Reviewer #1: (No Response)

Reviewer #2: Yes

3. Has the statistical analysis been performed appropriately and rigorously? 

Reviewer #1: (No Response)

Reviewer #2: Yes

4. Have the authors made all data underlying the findings in their manuscript fully available?

Reviewer #1: (No Response)

Reviewer #2: Yes

5. Is the manuscript presented in an intelligible fashion and written in standard English?

Reviewer #1: (No Response)

Reviewer #2: Yes

6. Review Comments to the Author

Reviewer #1: (No Response)

Reviewer #2: The author has answered all the reviewer's questions one by one and has made careful revisions to the manuscript, resulting in a significant improvement in the quality of the paper. The issues I was concerned about have been sincerely addressed by the author, and I have no further questions.

7. PLOS authors have the option to publish the peer review history of their article (what does this mean?). If published, this will include your full peer review and any attached files.

Reviewer #1: No

Reviewer #2: No

---

## [Editor Report · Acceptance letter]

22 Jun 2023

PONE-D-23-06470R1 

Tissue-Specific Reprogramming of Glutamine Metabolism Maintains Tolerance to Sepsis 

Dear Dr. Perry:

I'm pleased to inform you that your manuscript has been deemed suitable for publication in PLOS ONE. Congratulations! Your manuscript is now with our production department. 

Kind regards, 

on behalf of

Dr. Juan J Loor 

Academic Editor

PLOS ONE